# Optimization and Process Effect for Microalgae Carbon Dioxide Fixation Technology Applications Based on Carbon Capture: A Comprehensive Review

**Gang Li** [1], **Wenbo Xiao** [1], **Tenglun Yang** [1] **and Tao Lyu** [2,*]

[1] School of Artificial Intelligence, Beijing Technology and Business University, Beijing 100048, China; ligang@btbu.edu.cn (G.L.); crisp1110@163.com (W.X.); ytl1010@163.com (T.Y.)

[2] School of Water, Energy and Environment, Cranfield University, College Road, Cranfield, Bedfordshire MK43 0AL, UK

[*] Correspondence: t.lyu@cranfield.ac.uk

**Abstract:** Microalgae carbon dioxide ($CO_2$) fixation technology is among the effective ways of environmental protection and resource utilization, which can be combined with treatment of wastewater and flue gas, preparation of biofuels and other technologies, with high economic benefits. However, in industrial application, microalgae still have problems such as poor photosynthetic efficiency, high input cost and large capital investment. The technology of microalgae energy development and resource utilization needs to be further studied. Therefore, this work reviewed the mechanism of $CO_2$ fixation in microalgae. Improving the carbon sequestration capacity of microalgae by adjusting the parameters of their growth conditions (e.g., light, temperature, pH, nutrient elements, and $CO_2$ concentration) was briefly discussed. The strategies of random mutagenesis, adaptive laboratory evolution and genetic engineering were evaluated to screen microalgae with a high growth rate, strong tolerance, high $CO_2$ fixation efficiency and biomass. In addition, in order to better realize the industrialization of microalgae $CO_2$ fixation technology, the feasibility of combining flue gas and wastewater treatment and utilizing high-value-added products was analyzed. Considering the current challenges of microalgae $CO_2$ fixation technology, the application of microalgae $CO_2$ fixation technology in the above aspects is expected to establish a more optimized mechanism of microalgae carbon sequestration in the future. At the same time, it provides a solid foundation and a favorable basis for fully implementing sustainable development, steadily promoting the carbon peak and carbon neutrality, and realizing clean, green, low-carbon and efficient utilization of energy.

**Keywords:** microalgae technology; carbon dioxide fixation; economic benefit; emission reduction; mechanism

## 1. Introduction

With the abundant use of fossil fuels, the excessive emission of greenhouse gases, especially carbon dioxide ($CO_2$), has brought serious ecological damage and climate change to human society, and global warming has become a major problem to be solved urgently [1–3]. The impact of greenhouse gas accumulation is long term, with the average global temperature expected to rise by 2 °C by 2100 and 4.2 °C by 2400, which will cause great economic losses and be a global threat to food and nutrition security [4]. In China, the average crop yield loss is 2.58% with a temperature increase of 1 °C [5]. The $CO_2$ concentration increased to $2.46 \pm 0.26$ ppm/y in October 2021. If no measure is taken to control its development at the earliest convenience, the $CO_2$ concentration may reach 667 ppm in 2100 [6]. Therefore, global efforts are being made to mitigate climate change through reducing the $CO_2$ emission and/or enhancing carbon sequestration, which are crucial to long-term sustainable human development goals [7]. Reducing the use of fossil fuels is the straightforward way to reduce

$CO_2$ emissions. Further, a strategy of carbon capture, re-utilization, and storage to reduce $CO_2$ emissions is the most common technology for the moment [8].

Carbon capture and storage can meet the expanding worldwide energy demand by converting the captured $CO_2$ into fuels or using it to enhance the recovery of oil while dramatically lowering $CO_2$ emissions [9,10]. At present, common methods of $CO_2$ fixation include physical, chemical, and biological fixation methods [11]. For instance, the physical method is to inject $CO_2$ into the deep sea or geological layer to temporarily bury it, which cannot ensure that $CO_2$ will never escape due to the constraints of the geological environment, space, and cost [12]. The chemical method uses an adsorption material to directly fix or add alkaline neutralization reagent in the form of carbonate or bicarbonate to fix $CO_2$, which is relatively safe and permanent, but has the disadvantages of large reagent dosage and high cost [13,14]. The biological fixation method utilizes the photosynthesis of green plants to convert $CO_2$ into organic matter in the plant, provide energy sources, and maintain the C–O balance of the atmosphere. Of these, the biological approach has been deemed the most economically feasible, environmentally friendly, and sustainable method to capture and store carbon. Plants, microalgae, etc., are commonly used for biological carbon sequestration. In a general way, plants reduce carbon emission by only approximately 3–6%; but when grown under ideal conditions, microalgae such as cyanobacteria and green algae could be 50-fold more effective [15].

Yahya et al. studied the potential of microalgae as biological carbon fixers to support the development of a circular economy and environmentally sustainable coal-fired power plants [16]. By optimizing the culture parameter settings, Dasan et al. accelerated the $CO_2$ fixation efficiency of *Chlorella vulgaris* [17]. Premaratne et al. evaluated the $CO_2$ storage potential of *Desmodesmus* sp. in flue gas under nitrogen element limitation conditions, and applied the resultant biomass to biofuel preparation [18]. Ding et al. found that using native microalgae species to reduce effluents in palm oil mill and industrial discharge of $CO_2$ is an effective strategy [19]. To sum up, microalgae are a superior candidate for carbon capture and storage, which can also be utilized in biofuel production, wastewater treatment and other industries.

Microalgae with a simple structure, a short growth cycle, wide distribution, strong adaptability, and higher photosynthetic efficiency than conventional crops are mainly composed of eukaryotic and prokaryotic microalgae [20]. Microalgae directly use solar energy to fix $CO_2$ and produce $O_2$ and secondary metabolites, which is a typical representative of carbon sequestration organisms [11]. This process is also known as photosynthesis, including the primary reaction, electron transfer, photophosphorylation, and carbon assimilation (i.e., $CO_2$ fixation and formation of sugars). Because all crop biomasses obtain carbon from it, as shown in Figure 1, the Calvin–Benson cycle ($C_3$ cycle) plays a major role in the $CO_2$ fixation pathway in nature. Although land plants can also absorb $CO_2$ and produce organic matter through photosynthesis, microalgae can be planted in seawater or wastewater, can capture $CO_2$ from various sources, do not compete with traditional agriculture for soil resources, and achieve $CO_2$ fixation 10–50 fold more efficiently [21]. Every 1 g of biomass produced by microalgae can fix 1.83 g $CO_2$ [22]. Microalgae biomass with a high application value, which includes lipids, proteins, and polysaccharides, can be used as raw materials for biofuel, biochemical, food, medicine, and other industries. The species and environmental conditions of microalgae affect the photosynthesis and carbon sequestration efficiency of microalgae, which is beneficial to further effective utilization of microalgae [23].

In recent years, more and more reviews have focused on the influence factors and applications of microalgae $CO_2$ fixation technology, but there are few studies on the comprehensive analysis and discussion of microalgae based on the mechanism of carbon sequestration. Raeesossadati et al, mainly discussed the effect of important factors such as $CO_2$ concentration, photobioreactors, temperature, and light intensity on microalgae $CO_2$ fixation [24]. You et al. reviewed the application of wastewater treatment by microalgae [25]. By reason of the foregoing, based on previous studies, the optimization prospects

and process effect of microalgae $CO_2$ fixation technology are comprehensively reviewed as shown in Figure 2. Firstly, the mechanism of $CO_2$ fixation in microalgae is reviewed, and the main factors affecting carbon sequestration and growth are analyzed. Then, how to use strategies of random mutagenesis, adaptive laboratory evolution (ALE), genetic engineering to screen microalgae species that can improve the photosynthetic efficiency and the production of by-products is discussed. Moreover, combined with the current challenges faced by microalgae carbon sequestration technology, the application of microalgae carbon sequestration technology in $CO_2$ emission reduction, wastewater and flue gas treatment are analyzed. In the future, it is expected to establish a more optimized mechanism of microalgae carbon sequestration, which provides the reference for large-scale industrial application of $CO_2$ fixation in microalgae and the environmentally friendly economy in China.

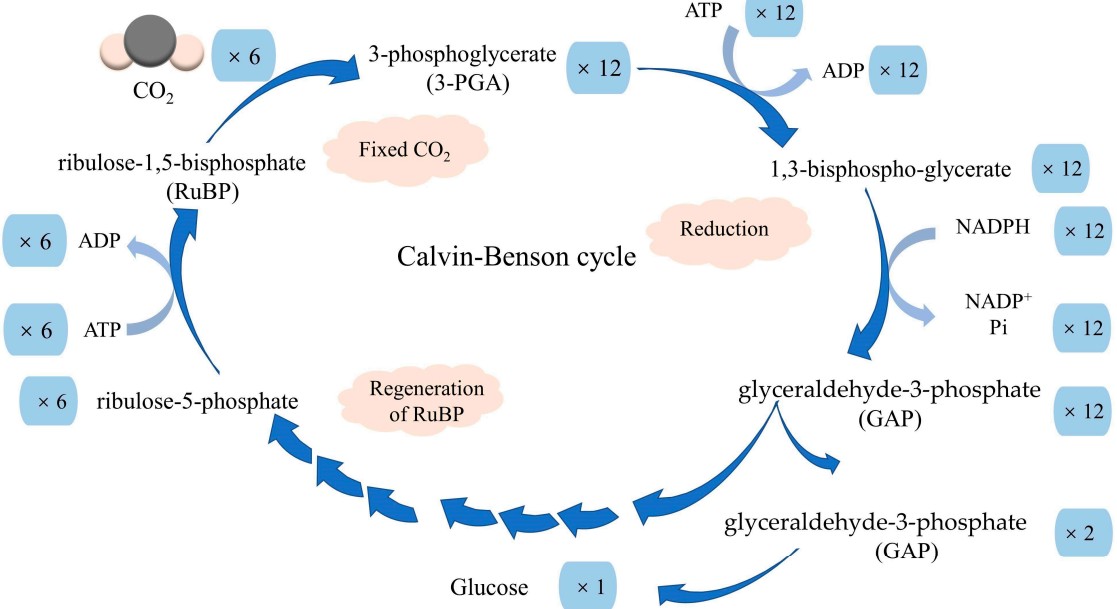

**Figure 1.** $C_3$ cycle.

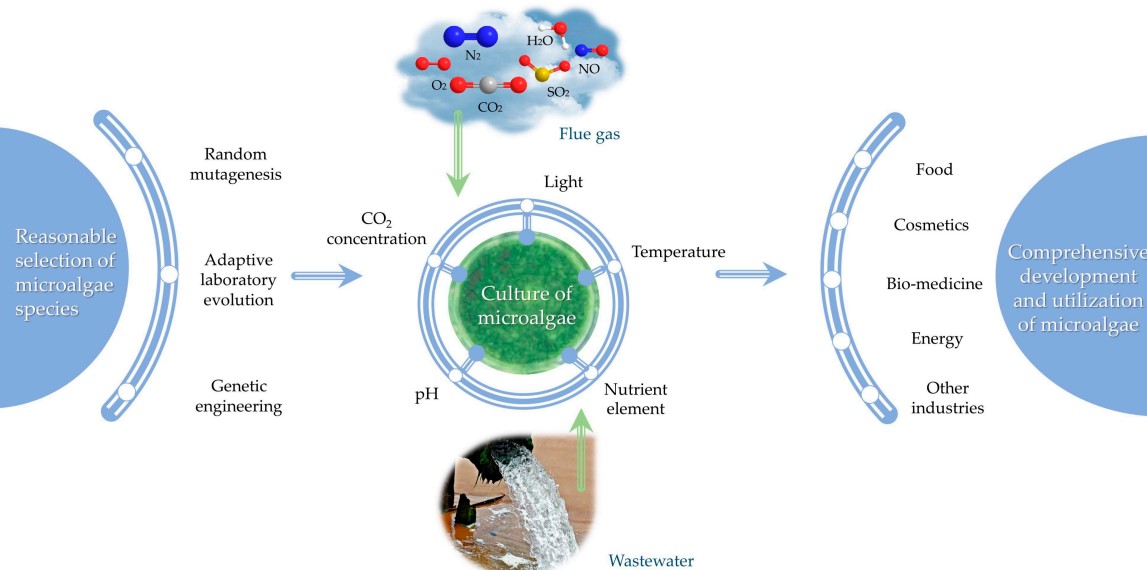

**Figure 2.** The technical flow chart of microalgae carbon sequestration.

## 2. Mechanism of Carbon Dioxide Fixation Technology by Microalgae

### 2.1. The Photosynthetic Carbon Metabolism Pathway

In essence, the photosynthesis of microalgae is a mechanism that allows them to utilize solar energy to exchange materials and energy with their surroundings and turn $CO_2$ into glucose [26]. It can be expressed by Equation (1):

$$CO_2 + H_2O \rightarrow (CH_2O)_n + O_2 \tag{1}$$

Photosynthesis is also divided into the photoreaction stage of converting the light energy into the active chemical energy and the dark reaction of converting the active chemical energy into the stable chemical energy. The dark reaction requires the strong reductant (NADPH) and the energy (ATP) generated by the light reaction to immobilize and reduce the $CO_2$ to sugar, as shown in Equations (2) and (3). The light energy is converted into chemical energy to resolve NADPH into $NADP^+$ through the photosystem of PSII and PSI. The pH gradient was produced on the thylakoid membrane and was applied to synthesize ATP [27].

$$H_2O + ADP + P_i + NADP^+ \overset{\text{light}}{\rightarrow} O_2 + ATP + NADPH + H^+ \tag{2}$$

$$CO_2 + ATP + NADPH + H^+ \rightarrow (CH_2O) + ADP + P_i + NADP^+ \tag{3}$$

Photosynthetic carbon metabolism in microalgae is mainly dependent on the $C_3$ cycle [28]. By employing ATP as an energy source, lowering the energy level, depleting NADPH, carbon enters the $C_3$ cycle as $CO_2$ and exits as sugar. This is what we see in Equation (4)—the $C_3$ cycle is simplified to carboxylation, reduction and regeneration of ribulose-1,5-bisphosphate (RuBP). RuBP entering the carboxysome is bound to $CO_2$ by the ribulose-1,5-bisphosphate carboxylase/oxygenase (rubisco) enzyme, which subsequently converts to a 2 molecules 3-phosphoglycerate (3-PGA) and diffuses from the carboxysome to the cytoplasm through the pores of the hexamer shell protein. Equation (5) shows the reduction phase, where 3-PGA in the cytoplasm forms glyceraldehyde-3-phosphate (GAP) catalyzed by the enzyme. The regeneration of RuBP ensures the continuous operation of the carbon sequestration cycle.

$$3RuBP + 3CO_2 \rightarrow GAP + 3RuBP \tag{4}$$

$$PGA + ATP + NADPH + H^+ \rightarrow GAP + ADP + NADP^+ + P_i \tag{5}$$

The rubisco has the dual functions of oxygenation and carboxylation, specifically which the choice of function is influenced by $CO_2$ and $O_2$ concentrations. Due to the relatively high concentration of $O_2$ in the atmosphere, it is conducive to the work of oxygenase and promotes photorespiration, which leads to the reduction in photosynthesis production. Therefore, microalgae have developed $CO_2$-concentrating mechanism (CCM) to maximize photosynthetic efficiency under low $CO_2$ concentrations or inorganic carbon (Ci) conditions [29].

### 2.2. Carbon Dioxide Concentrating-Mechanism

When $CO_2$ is dissolved in fluid, it exists in four specific Ci forms: $HCO_3^-$, $CO_3^{2-}$, $H_2CO_3$, and dissolved $CO_2$. The proportion of Ci varies with pH. It's possible that different microalgae strains prefer various Ci types. For instance, *Nannochloropsis oculata can* only actively transport $HCO_3^-$, and *Chlamydomonas reinhardtii* be able to use simultaneously $CO_2$ and $HCO_3^-$ [30]. In truth, pH 7.5–8.4 is the typical range for the culture medium used by microalgae. Approximately 90% is $HCO_3^-$ in this culture medium, and the concentrations of $CO_3^{2-}$ and $H_2CO_3$ are low [31]. In the process of acclimatizing to the concentration of Ci changes in the environment, the microalgae have formed CCM that can help to adapt to the changes in the external $CO_2$ concentration. CCM is based on a single cell, which can actively transport and accumulate Ci at low $CO_2$ levels by counteracting

the inefficiency of rubisco to boost photosynthetic efficiency. It is found in almost all eukaryotic microalgae and cyanobacteria and plays a crucial role in the fixation process of carbon [32]. In both eukaryotic and prokaryotic, the CCM of microalgae includes three main systems: (1) Ci transporter; (2) Carbonic anhydrase (CA) that is used to convert Ci to $CO_2$; (3) microcompartments with rubisco for the delivery of $CO_2$ [29]. Microalgae concentrate the Ci pumped into the cell at the photosynthetic site to form high carbon sites nearly a 1000-fold higher than the surroundings, and convert them into organic carbon through the rubisco to achieve the carbon fixation [28]. This is also regarded as the rate-limiting step of the entire cycle. Rubisco in the pyrenoids of eukaryotic algae or in the carboxysomes of cyanobacteria has a low affinity for $CO_2$ and requires higher concentrations of $CO_2$ to obtain a normal rate of reaction [33].

The simplified model of CCM is shown in Figure 3, which is mainly divided into two stages: The first stage involves obtaining Ci from the environment and delivering $CO_2$ and $HCO_3^-$ to the chloroplast; in the second stage, Ci crosses the thylakoid membrane and is reduced to $CO_2$ by CA at higher ambient pH, which increases the concentration of $CO_2$ near the rubisco and ultimately enhances the photosynthetic rate [32]. $HCO_3^-$ can enter cells through active transport, while $CO_2$ can enter cells through inactive diffusion in microalgae. CA regulates $CO_2$ and $HCO_3^-$ to maintain proper pH in the chloroplast stroma. maintains, so CA has a noticeable effect on catalyzing $CO_2$ conversion [31]. In addition, the activity of rubisco, the pH of chloroplast stroma, etc., can also affect the $CO_2$ conversion efficiency [34]. The favored form of stacking Ci is $HCO_3^-$ that is approximately 1000-fold less permeable to lipid membranes than the uncharged $CO_2$ molecules. The $CO_2$ absorption system can recycle or recapture leaking $CO_2$ in order to efficiently fix the $CO_2$ and prevent its escape from the cell [35].

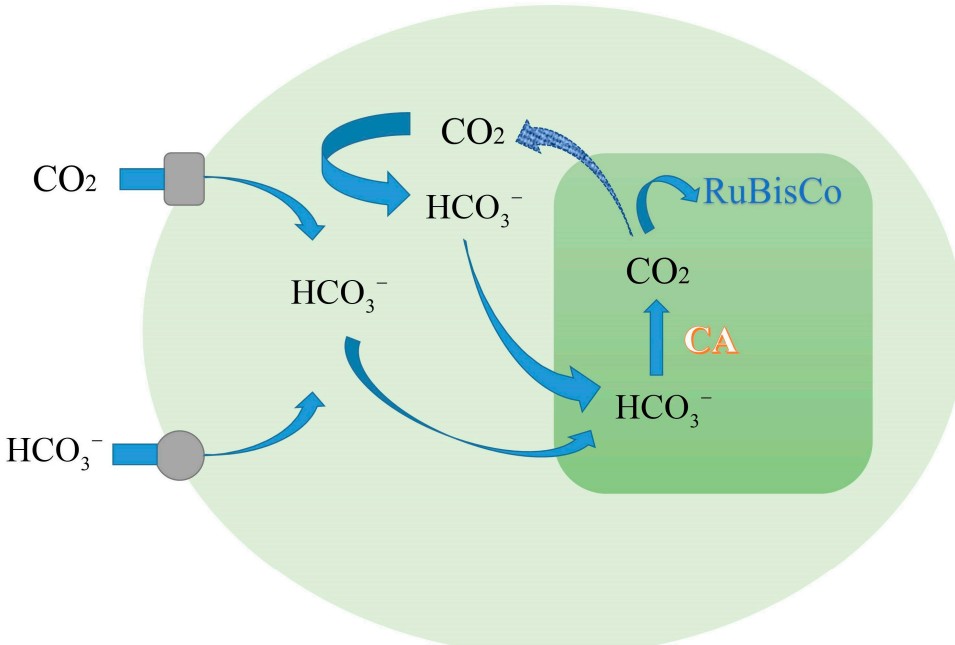

**Figure 3.** The simplified model of CCM.

## 3. Environmental Conditions for the Growth of the Microalgae

The effect of diverse environmental conditions on the growth rate of microalgae is depicted in Figure 4. The optimal range of conditions to achieve high $CO_2$ fixation rates and the growth efficiency also vary by microalgae species. The effects of microalgae growth conditions on carbon sequestration efficiency and biomass production are summarized below.

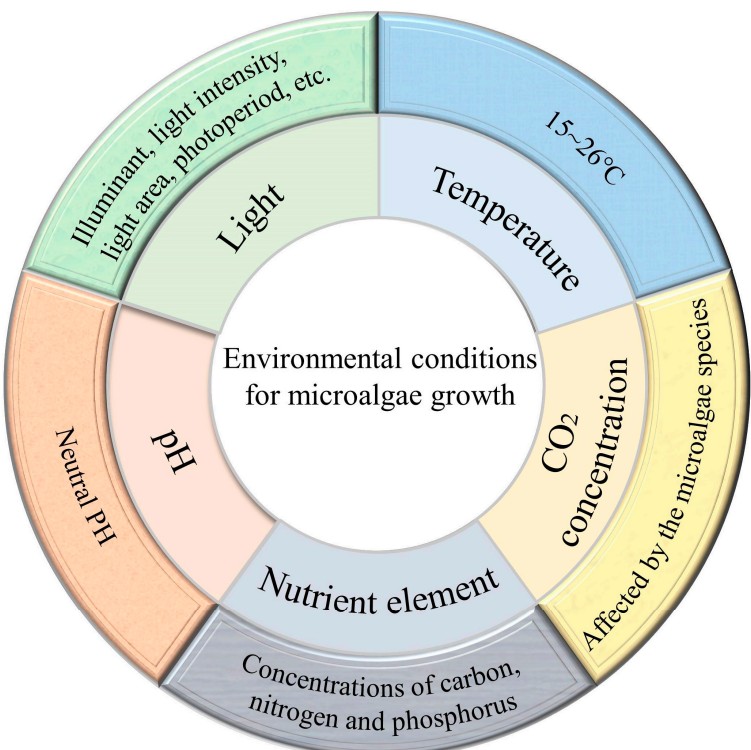

**Figure 4.** Different growth environment conditions affect the growth rate of microalgae.

### 3.1. Light

Light is the basic energy source affecting the photosynthesis dynamics of microalgae, which can affect cell growth and metabolism by controlling the light source, light intensity, interval, area, etc. [36]. Excessive light intensity may lead to photooxidation and photoinhibition, while low light levels would limit growth [37]. The majority view is that photoinhibition is the primary factor reducing algal output. Until light saturation is attained, the microalgae growth rate increases linearly with increasing light intensity at low light levels. The optimal light intensity for microalgae growth is usually 26–400 μmol photons m$^{-2}$ s$^{-1}$ [38]. Lipid synthesis of *tetradesmus obliquus* reached the maximum at 200 μmol m$^{-2}$ s$^{-1}$, accounting for 45.31% of the dry biomass, which would decrease provided that further augmented the light intensity [39]. Additionally, according to Kim et al., microalgae production rates of *Scenedesmus* sp. was about 45% higher than at a single wavelength at 400–700 nm white light [40]. By controlling the light wavelength, photosynthesis, gene transcription, enzyme activation and cellular composition can be impacted to the point that the biomass and lipid content of microalgae are altered [41].

### 3.2. Temperature

Temperature is a major factor regulating the cellular, morphological, and physiological responses of microalgae, now that it impacts their photosynthesis, which leads to changes in the carbon sequestration efficiency of microalgae. Zhao et al. revealed that the most suitable temperature for the growth of the most common microalgae was between 15 and 26 °C [42]. Overall, with decreasing temperature within the appropriate range, rubisco activity reduces, the index of unsaturated fatty acid augments, the growth rate reduces, biomass is reduced, and carbon fixation efficiency also reduces significantly. Nevertheless, high temperatures frequently have irreversible damage on microalgae. Sachdeva et al. discovered that *Chlorella pyrenoidosa* M18 were able to survive at temperatures up to 47 °C, with the highest average growth rate at 37 °C. At high temperature or direct sun, this algal species had a higher lipid yield of 44.51%, compared to room temperature [43]. As shown in Table 1, the optimal growth temperature of microalgae varies by species,

and is also influenced by other environmental parameters (e.g., light intensity and $CO_2$ concentration) [44].

**Table 1.** The optimal growth temperature for different species of microalgae.

| Species | Culture Medium | Optimal Growth Temperature (°C) | Average Specific Growth ($d^{-1}$) | Ref. |
|---|---|---|---|---|
| *Chlorella pyrenoidosa* M18 | BG11 | 37 | 0.70 | [43] |
| *Thermosynechococcus elongatus* PKUAC-SCTE542 | BG11 | 55 | 0.22 | [45] |
| *Chlorogleopsis* sp. | BG11 | 50 | 0.14 | [46] |
| *Chlorella* sp. MT-15 | Artificial sea water | 30 | Approximately 0.80 | [47] |
| *Chlorella* sp. MT-7 | Artificial sea water | 30 | Approximately 0.60 | |
| *Thermosynechococcus* sp. CL-1 | Modified Fitzgerald | 50 | 2.70 | [48] |
| *Nannochloropsis* sp. Oculta | Modified Fitzgerald | 30 | 1.60 | |

*3.3. pH*

By altering the activity of the cellular metabolic enzymes and the uptake and usage of ions by microalgae cells, the pH value influences the physiological metabolism of microalgae. The energy costs of $HCO_3^-$ transfer to cytosol are decreased at pH ranges between 7.5 and 8.5, which lowers the cost of carbon fixation [49]. The depletion of Ci by microalgae cells growth leads to an increase in pH, which might alter the biochemical reaction properties of microalgae, in turn with promoting cell rupture. Changes in pH can affect CA activity, leading to affect microalgae growth. Buffers such as sodium hydroxide or calcium carbonate to adjust pH to the best level adapted to microalgae growth can further boost the fixation rate of $CO_2$ and biomass production [50].

Most microalgae are suitable for cultivation under neutral pH conditions [50], with exceptions, such as *Chlorococcum* could live in pH 4.0, and *Spirulina* at pH 11.0 [51]. Razzak et al. discovered that *Nannochloropsis oculata* grew well between medium pH 5.5 and 6.5 [51]. *Graesiella* sp. WBG-1 had the highest $CO_2$ fixation rate and lipid content at pH 8.0–9.0, which were 0.26 g/L/d and 46.28%, respectively. Although the usage rate of $CO_2$ increased along with the pH, the optimum pH for microalgae development did not have the highest utilization rate of $CO_2$ [52]. Therefore, it is essential to cultivate algae species that can grow well under high pH values in order to effectively utilize the carbon sequestration capacity of microalgae.

*3.4. Nutrient Element*

The building blocks of microalgae's cell synthesis include carbon, nitrogen, and phosphorus, which are also crucial nutrients for the biomass growth. To some extent, the photosynthesis of microalgae is influenced by the species, morphology, and quality of the nutrients. Carbon–nitrogen ratio (C/N) is among the important factors affecting carbon fixation efficiency, biomass accumulation and productivity of value-added components. *Spirulina platensis* had been nuclear radiated and cultured in specific cultures. When the ratio of $NH_4HCO_3$ and $NaNO_3$ was set at 1:4, carbon utilization efficiency of the hybrid process could reach 40.45%, But the excessively high concentration of $NH_4HCO_3$ will also generate toxicity for microalgae, which results in a lower biological yield [53]. Appropriately increasing the concentration of phosphorus is beneficial to microalgae development and lipid accumulation because some microalgae may utilize it to manufacture organic esters [54,55]. Microalgae cause phosphorus to convert ADP into ATP through phosphorylation, which can also precipitate phosphate through cell adsorption or regulating pH as shown in Figure 5. Because wastewater contains the above elements, growing microalgae can remove pollutants while fixing carbon, but this also faces some challenges, as detailed in a later article [30].

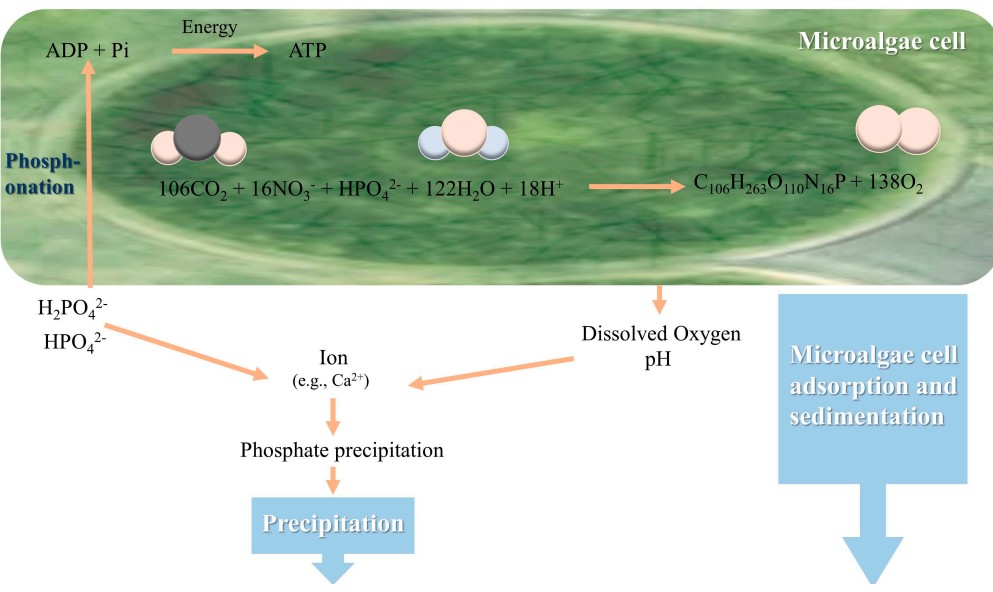

**Figure 5.** Mechanism of adsorption of settling phosphorus by microalgae.

### 3.5. Carbon Dioxide Concentration

The capability of different microalgae to tolerate $CO_2$ is different. Even though certain microalgae have great $CO_2$ tolerance and can grow across most of the $CO_2$ concentration range, the optimal growth concentration is determined. The statistics are shown in Table 2 below. Some microalgae can grow normally at low concentrations of $CO_2$, while some will only show high growth rates at high concentrations of $CO_2$. Either a too high or too low $CO_2$ concentration reduces the $CO_2$ fixation efficiency and the biomass yield. To some degree, increasing the $CO_2$ concentration can enhance the carboxylation activity of the rubisco, suppress its oxidative activity, and accelerate photosynthesis [56]. However, too high a $CO_2$ concentration will lower pH, which will reduce the activity of CA and impede cell growth [57].

**Table 2.** Growth rates of different species of microalgae at different $CO_2$ concentrations.

| Species | $CO_2$ (%) | Culture Medium | Aeration Rate (vvm) | $CO_2$ Fixation Rate (mg/L/d) | Biomass Yield (g/L) | Biomass Productivity (mg/L/d) | Ref. |
|---|---|---|---|---|---|---|---|
| *Chlorella vulgaris* | 2% constant 4% provided intermittently | Liquid medium of 3N-BBM+V | 0.4 | 4110 4500 | 2.59 2.62 | 3530 3410 | [58] |
| *Spirulina platensis* | 5 | Zarrouk medium | 0.1 | 178.46 | 1.75 | - | [43] |
| *Dunaliella* sp. ABRIINW-SH33 | 10 20 30 | Modified Johnson medium | 3 | 455.74 317.32 279.74 | 2.98 2.08 1.83 | 248.60 173.08 152.60 | [59] |
| *Dunaliella* sp. ABRIINW-CH2 | 10 20 30 | | | 423.19 317.32 272.56 | 2.77 2.08 1.78 | 230.83 173.08 148.67 | |
| *Chlorella vulgaris* FACHB-31 | 15 | Modified BG11 | 0.02 | 878.40 | 3.35 | - | [60] |
| *Botryococcus braunii* | 0.03 10 20 | BG11 | 0.1 | - | 0.64 0.41 0.26 | - | [61] |
| *Scenedesmus* sp. | 0.03 10 20 | BG11 | 0.1 | - | 0.72 0.90 1.90 | - | |
| *Anabaena* sp. CH1 | 10 | Arnon medium | 0.4 | 1010 | 1.16 | - | [62] |
| *Scenedesmus obliquus* | 5 | Selenite enrichment medium | - | 577.60 | - | - | [63] |
| *Chlorella protothecoides* | 20 | BG11 | - | 370 | 1.55 | 190 | [64] |
| *Chlorella vulgaris* P12 | 6.50 | - | 0.5 | 2290 | 9.97 | 1330 | [56] |

**Table 2.** *Cont.*

| Species | CO$_2$ (%) | Culture Medium | Aeration Rate (vvm) | CO$_2$ Fixation Rate (mg/L/d) | Biomass Yield (g/L) | Biomass Productivity (mg/L/d) | Ref. |
|---|---|---|---|---|---|---|---|
| *Chlorella* sp. L38 | 5 | BG11 | - | - | 0.60 | - | [65] |
| *Spirulina* sp. LEB 18 | 10 | Zarrouk medium | - | 160 | 1.07 | 20 | [66] |
| *Scenedesmus obliquus* (FACHB-13) | 15 | BG11 | 1 | - | 1.61 | - | [67] |
| *Chlorella vulgaris* | 15 | BG11 | - | 120 | 1.83 | 144 | [68] |
| *Scenedesmus obliquus* UTEX 393 | 5 | Airlift photobioreactor | 0.43 | 727.70 | - | 405.70 | [69] |
| *Chlorella* sp. AE 10 | 30% | BG11 | - | - | 3.68 | - | [70] |

The maximum biomass output and the CO$_2$ fixation rate of *chlorella vulgaris* both increased by 57% and 56%, respectively, as the CO$_2$ content increased from 3% to 7% [71]. Rodas-Zuluaga et al. indicated that when growing microalgal strains at 0.03% to 20% CO$_2$, *Botryococcus braunii* grew well at CO$_2$ of 0.03%, while *Scenedesmus* sp. had the highest biomass yield at 20% CO$_2$ [61]. These findings all point to the influence of microalgal species on the growth and CO$_2$ tolerance of microalgae.

### 4. Strategies for Improving Photosynthetic Efficiency in Microalgae

Selection of microalgae strains with a fast growth rate, strong environmental adaptability, and high CO$_2$ fixation ability is the most important step to improve the effect of CO$_2$ fixation in microalgae. In addition to selecting microalgae with excellent traits from natural strains, random mutagenesis, ALE, and genetic engineering that enhancing the specific metabolic phenotype of microalgae and improving their environmental adaptability are also commonly used strategies, as shown in Figure 6. Moreover, the maximum productivity of target compounds and conditions for rapid growth of microalgae are often mutually exclusive, and algal productivity can be further enhanced by means such as mutagenesis and/or genetic engineering [72]. The effects of these methods on improving the growth performance of microalgae are shown in Table 3.

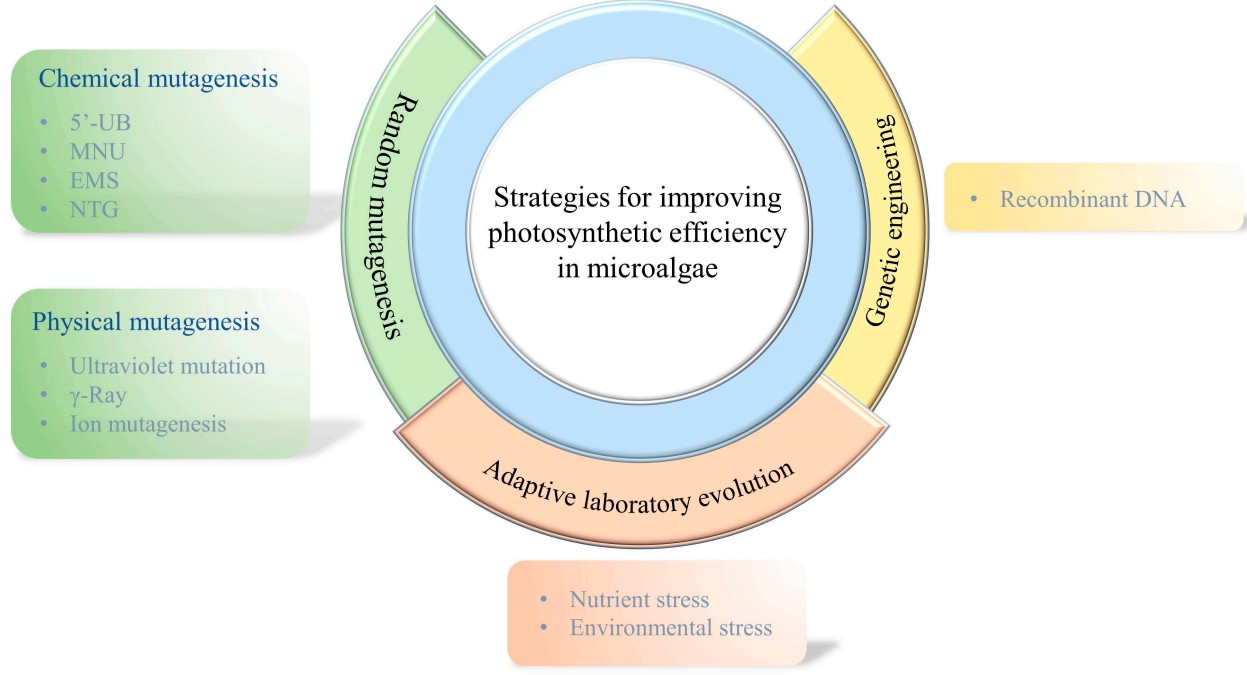

**Figure 6.** Strategies for improving photosynthetic efficiency in microalgae.

**Table 3.** Application of strategies to improve the photosynthetic efficiency of microalgae.

| Strategies | | Organism | Phenotype | Ref. |
|---|---|---|---|---|
| Random mutagenesis | Chemical mutagenesis | *Chlorococcum* sp. FFG039 | The FFG039 PM exhibited 1.7-fold and 1.9-fold higher biomass and lipid productivities than those of the wild type. | [73] |
| | | *Chlorella* sp. | The E100-30-60 showed that the highest biomass yield and biomass productivity were 111 and 110% higher than the wild type, respectively. | [74] |
| | Physical mutagenesis | *Haematococcus pluvialis* | The average specific growth rate of *Haematococcus pluvialis* mutated with 4000 Gy γ-ray irradiation was increased by 15% compared with the original strain with air aeration. | [75] |
| | | *Chlorella vulgaris* | The resulting mutant resulted in a 33% increase in lipid yield. | [76] |
| ALE | | *Chlorella* sp. | The maximal biomass concentration of AE10 was $3.68 \pm 0.08$ g/L in 30% $CO_2$, which was 2.94 fold compared to the original strain. | [70] |
| | | *Crypthecodinium cohnii* ATCC 30556 | The cell growth of the evolved strain (FS280) was increased by 161.87%. | [77] |
| | | *Nannochloropsis oceanica* | The growth rate of mutants was enhanced by 32%, and biomass accumulation by 46%. | [78] |
| Genetic engineering | | *Chlamydomonas reinhardtii* | The strain improved photosynthetic productivity. | [79] |

*4.1. Random Mutagenesis*

Random mutagenesis with simplicity of operator, high reliability, and non-orientation can be applied to culture mutants with the optimal features of a high carbon fixation efficiency, a high lipid yield, and tolerated $CO_2$ [80]. Random mutagenesis applied to any microalgae strain can be divided into two categories—chemical mutagenesis and physical mutagenesis, using physical or chemical means to influence the strain—resulting in random alterations in the genome. Its advantage is that this technology can mutate and ameliorate various microalgae traits without requiring comprehending the complex knowledge of physiology or genetics, and the mutants produced are not genetically modified organisms [72,81].

4.1.1. Chemical Mutagenesis

Chemical mutagenesis is for the most part used as mutagens by ethyl methane sulfonate (EMS), 1-methyl-3-nitro-1-nitrosoguanidine (MNNG), N-methyl-N-nitrosourea (MNU), N-methyl-N′-nitro-N-nitrosoguanidine (NTG), 5-bromouracil (5′-UB), etc. [82]. The advantage of chemical mutagenesis is that it does not need special equipment, can produce a high frequency of point mutations, create relatively few chromosomal aberrations, but most of the mutagens are toxic or carcinogenic, easily pollute the environment [80]. Tanadul et al. showed that the strain E200-30-40, after EMS mutagenesis, produced 59% and 53% higher lipid content and productivity than the wild-type strain, respectively, which increased biomass and lipid accumulation [74]. Nojima et al. dispersed the cells of water surface-floating microalgae strains (*Botryosphaerella* sp. AVFF007 and *Chlorococcum* sp. FFG039) treated with EMS or MNNG mutagen to obtain mutant FFG039 PM, with a biomass and lipid productivity of 1.7- and 1.9-fold greater than the wild-type strains in the absence of inhibiting biofilm formation and floating capacity [73]. The total production of carotenoids and astaxanthin in the mutant G1-C1 of *Coelastrum* sp. after selection by EMS and glyphosate was approximately double that of the wild type [81]. As can be seen, chemical mutagenesis is utilized frequently due to simplicity of operation and potent mutagenicity.

### 4.1.2. Physical Mutagenesis

Physical mutagenesis mainly includes ultraviolet (UV) mutagenesis, $\gamma$-ray mutagenesis, and ion mutagenesis. UV mutagenesis is the most common because its control is flexible, economical, fast, and effective. Compared with chemical mutagens, it does not lead to poisoning of operators and has no secondary contamination [80]. During operation, some precautions also need to be taken to reduce the risk of UV radiation to the experimenter. The mutation effect depends on microalgae characteristics and experimental conditions, and is not applicable to all species [83]. Moha-León et al. found that using UV radiation and the herbicide quizalofop-p-ethyl to select freshwater microalgae, the chosen strains had more fatty acids and fat in them [84]. The screened mutant lipid production soared by 33% using UV irradiation of *Desmodesmus armatus* and *Chlorella vulgaris* [76]. $\gamma$-rays have more deep penetration than UV, which can penetrate cells to enhance the expression of photosynthesis enzymes (e.g., rubisco), rearrange chromosomes, screen oxygen radicals, and ultimately alter genetic traits [75]. The biomass of the $\gamma$-rays domesticated mutant *Chlorella* PY-ZU1 was increased 2.3-fold relative to the original strain [85]. Numerous studies have proved that physical mutagenesis changes the physiological and biochemical characteristics that are typical of cells. As a result, new strains with much higher lipid enrichment capacities can be chosen.

### 4.2. Adaptive Laboratory Evolution

ALE is a strain improvement method based on random mutation and natural selection that can be used as a tool to study evolution, which improves the phenotype, performance, and stability of microalgae, divided into intermittent and continuous culture patterns [86,87]. Additionally, it compensates for the neglect of molecular genetic mechanisms in Darwinian evolution and its development, using high-throughput DNA sequencing as a tool to effectively model the evolutionary process of selection [88]. The efficiency of ALE rests with the original strain, the initial cell density of the microalgae, and the strategy of stress [89]. Nutritional stress and environmental stress are the two types of stress. Nutritional pressure is often used to improve the completion rate of some low-cost substrates, and environmental pressure mainly includes temperature, oxidation, and organic solvent tolerance. [90]. Choosing the appropriate pressure can effectively improve the efficiency of ALE to a certain extent. Under laboratory control conditions, ALE cultures of microorganisms reach certain targets and subculture until the metabolic phenotype is stable and the evolved microalgae strains containing beneficial mutations are obtained.

In the ALE of microalgae, a single stress is typically used to better appreciate the process of tolerance [89]. Using reasonable ALE can not only improve the tolerance of microalgae to abiotic stress, but also optimize the yield of target metabolites and enhance the utilization of pollutants [86]. Controlling light factors by ALE could increase the microalgae growth rate and the accumulation of $\beta$-carotene and lutein. Thus, light stress can accelerate the removal of nitrogen and phosphate velocities in wastewater treatment [88]. At 10% and 20% $CO_2$, *Chlorella* sp. strains AE10 and AE20 were obtained by ALE, and had higher $CO_2$ tolerance than *Chlorella* sp. [70]. According to Table A1, ALE is a feasible strain enhancement tactic that can take advantage of microalgae's biotechnological potential to boost the ability of carbon fixation, lipid production, and the biomass concentration for distinct strains.

### 4.3. Genetic Engineering

Genetic engineering focuses on the gene expression and transcription, altering the target genome by using recombinant DNA at the molecular level. In principle, there are three pathways: (1) to provide the necessary phenotype, add heterologous genes; (2) to stop gene expression through gene disruption or gene deletion, or lower the level of existing genes' expression through RNA interference; (3) to enhance and induce expression under non-native promoters. Despite the complexity of the process, it is beneficial to boost photosynthesis, increase microalgal biomass, and increase $CO_2$ absorption rates [72]. The

biomass productivity of *Nannochloropsis oceanica* can be elevated by overexpression of a nuclear-encoded. At air $CO_2$ concentrations, the mutant growth rate increased by 32% and biomass accumulation increased by 46% [78]. Genetic engineering can decrease the uncontrollability and blindness of microalgae mutagenesis, but not all species of microalgae are suitable for genetic engineering, and the resulting transgenic organisms are still seen as potential risks to the environment and human health. The foremost limitations of genetic engineering are that it is time-consuming, costly, and requires complex equipment and experimental conditions.

## 5. Application of Carbon Dioxide Fixation Technology in Microalgae

### 5.1. In the Atmosphere

As mentioned above, microalgae cultures are considered a promising strategy to capture $CO_2$ from the atmosphere due to their high growth rate and $CO_2$ fixation capacity. In December 2022, the $CO_2$ levels in the atmosphere reached 418 ppm [91]. Microalgae use CCM to increase the $CO_2$ concentration near rubisco in order to achieve carbon fixation [92]. Tsai et al. proved that the symbiosis of microalgae with a natural medium has maximum $CO_2$ consumption efficiency and favorable $CO_2$ fixation ability [93]. Microalgae are typically cultivated in enclosed systems or open ponds, where they can absorb $CO_2$ from the air to supply the cell growth and eventually produce lipids, carbohydrates, and proteins [94,95]. Open ponds are often used in large-scale industrial growth systems to grow microalgae because they are cost-effective and have a higher production capacity compared to closed systems (i.e., photobioreactor). However, disadvantages include the large area, unstable culture conditions, easy pollution, and that they are limited by too few microalgae strains. A photobioreactor can achieve aseptic operation, have a relatively stable culture conditions, and prevent the evaporation of water, but its construction and operation costs are higher. Although direct air capture requires no equipment and energy, the effectiveness of carbon sequestration is relatively low and is affected by environmental factors [30,96,97]. Therefore, the common microalgae culture systems are generally open ponds and photobioreactors.

### 5.2. Flue Gas

Flue gas is a cheap and abundant source of $CO_2$, coal flue gas contains a 12–15 volume percentage (vol%) of $CO_2$, while natural gas contains a 4–8 volume percentage (vol%) of $CO_2$ [98,99]. The composition of flue gas is more complex, and will depend on the type of combustion raw materials, generally containing a large proportion of $N_2$ and $CO_2$, a little bit of $H_2O$ and $O_2$, and a minor amount of nitrogen and sulfur oxides, heavy metals, dust, and other contaminants. The use of microalgae to reduce $CO_2$ emissions in flue gas is very promising, but the high concentration of some pollutants (e.g., $SO_X$ and $NO_X$) may have severe toxic effects on microalgae cells and inhibit the growth of microalgae, as well as inhibit the ability of microalgae to sequester carbon, as shown in Figure 7 [100]. Among the four flue gas microalgae remediation strategies of microalgae, the use of microalgae to directly mitigate the flue gas pollution method has higher economic and environmental benefits because it does not use any adsorbent and can use cultured microalgae to produce secondary products [92]. The associated expenses, pollutant emissions, and energy requirements are lowered since the $CO_2$ concentration for algae development is comparable to the average flue gas $CO_2$ concentration [101]. Therefore, strain screening that can tolerate a high $CO_2$ content and high temperature resistance is the premise of realizing $CO_2$ fixation in flue gas.

The reduced growth rate at high $CO_2$ concentrations is often associated with acidification, leading to inactivation of key enzymes in the $C_3$ cycle, and homeostatic reactions enable $CO_2$-resistant strains to adjust their pH [102]. Yen et al. showed that *Chlorella* sp. achieved the maximum growth rate at a 10% $CO_2$ concentration, which shows a high $CO_2$ tolerance [103]. *Chlorella fusca* LEB 111 exhibited the largest daily bio-fixation amount of $CO_2$ in flue gas [104]. Microalgae cells can capture low levels of nitrogen oxides as sources of nitrogen. Further, there are polysaccharides, proteins, and lipids on the surface of the cell

wall of microalgae, which contain charged functional groups that can attract and combine with heavy metals. Consequently, microalgae can be employed as adsorbents to effectively remove heavy metals [103].

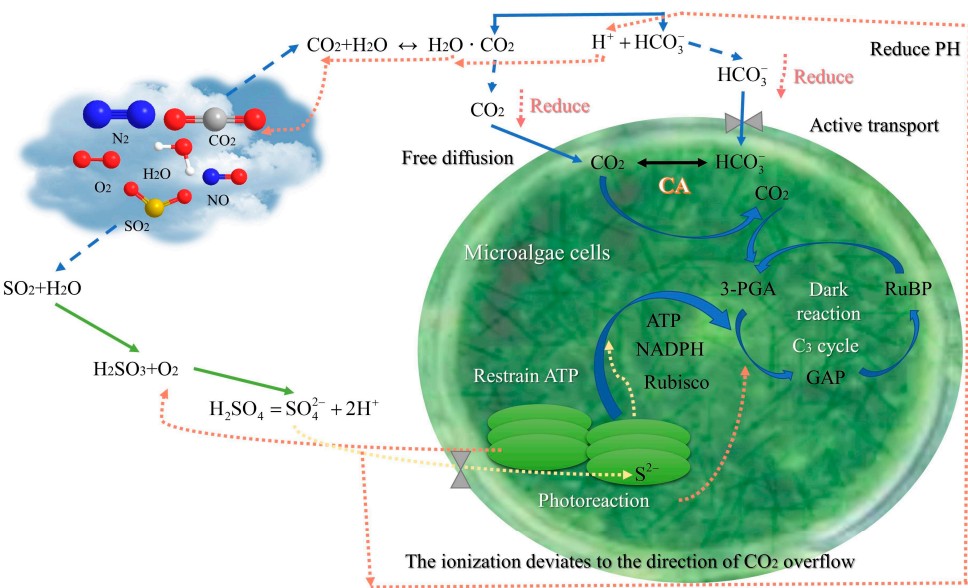

**Figure 7.** Mechanism of the inhibition of microalgae $CO_2$ fixation by $SO_X$.

In conclusion, the use of microalgae to capture $CO_2$ from flue gas, while removing the nitrogen oxides, sulfur oxides, and heavy metals in flue gas is an efficient way to reduce $CO_2$ emissions and air pollution. Accordingly, obtaining sufficient carbon sources contributes to the accumulation of biomass lipids and the increase in biological production, which can be applied to raw materials for biofuels or biorefinery. This approach maximizes economic profit, ensures the sustainable and ecological benign production of biomass, and is consistent with the development requirements of decreasing $CO_2$ emissions, which mitigates the effect of human activity on the climate. For now, most of the research on flue gas still has attached importance to laboratory research under controlled conditions, so further research and optimization are essential.

*5.3. In Wastewater*

The increasing urbanization and growth of the world's population have resulted in a steep rise in water consumption, making wastewater treatment among the greatest difficulties in the world. Microalgae wastewater treatment systems have a substantial economic value compared to conventional wastewater treatment technologies, which rely on expensive energy and chemicals and increase greenhouse gas emissions [105]. Wastewater contains most of the nutrients needed for microalgae growth and the trace metals necessary for photosynthesis, such as ammonias, phosphates, irons, coppers, and zincs. The combination of wastewater purification, $CO_2$ capture, and microalgae cultivation can reduce the cost of sewage treatment, lower the use of chemicals in the processing process, and slow down the emission of $CO_2$. Additionally, the resulting biomass can be further processed to produce biofuels, which alleviates the current energy crisis [98,106].

The efficiency of wastewater treatment by microalgae is relevant to wastewater type, nutritional characteristics, photobioreactor types, and culture methods [25]. The tolerance of organic pollutants in wastewater also varies among various species, as shown in Table A2. The content of nitrogen and phosphorus in domestic wastewater is low, the concentration of toxic and harmful substances in agricultural wastewater is low, and industrial wastewater usually contains a large amount of oil, heavy metals, and toxic pollutants. Song et al. showed that *Chlorella* sp. was cultivated in a system mixed with soybean wastewater and bicarbonate solution ($NH_4HCO_3$). At pH = 7, the low-$NH_4HCO_3$ concentration system had

the highest biomass yield (0.74 g/L) and the highest carbon bioconversion efficiency [107]. *Scenedesmus dimorphus* was cultured in tertiary urban sewage with different NP ratios, and when the NP ratio was 8:1 at 4% $CO_2$, the maximum biological fixation rate of microalgae reached 49.6 mg/L/d [108]. Hu et al. proved that *Chlorella sp.* L166 realized the efficiency of a maximum removal of TN, TP, COD, and $CO_2$ with an NP ratio of 5:1 in 20% soybean wastewater, introducing 5% $CO_2$ at a rate of 0.1 vvm [109].

Combining the capture of $CO_2$ from flue gas with wastewater treatment using microalgae culture to produce biofuels is a sustainable and economical wastewater treatment technology [110]. The extraction of high-value-added products in microalgae will also immensely encourage its large-scale industrialization that has a wide range of application potential.

### 5.4. Other Applications

Microalgae contain high-value fatty acids, pigments, and vitamins, etc. After special treatment, microalgae biomass can also be converted into biodiesel, methane, and other energy sources. Microalgae with a high oil content have the potential to produce 25-fold higher oil yields than conventional biodiesel crops such as the oil palm [111]. Astaxanthin extracted from *Haematococcus pluvialis* is often used in cosmetics as to its strong antioxidant capacity. In addition, eating astaxanthin is thought to prevent or control different diseases, and it is commonly used as a red pigment in food [112]. Polyunsaturated fatty acid has conducive effects on fetal development, preventing cardiovascular disease, and even improving cognitive function in Alzheimer's disease patients, which is used as nutritional supplements [113]. In conclusion, microalgae biomass can be applied in food, bio-medicine, cosmetics, energy, and other industries, with a high potential application value.

## 6. A Future Prospect for Microalgae Carbon Dioxide Fixation Technology

### 6.1. Selection of Seeds Rationally

Different microalgae have different carbon fixation effects, so microalgae species that have a fast growth rate, strong environmental adaptability, high photosynthetic efficiency, a high biological yield, and are easy to cultivate outdoors on a large scale can be selected in order to improve the photosynthesis efficiency of microalgae [114]. In the process of industrial application, due to the differences in actual conditions, natural strains sometimes cannot meet the production requirements. The measures of random mutagenesis, ALE, and genetic engineering are adopted to obtain more evolved microalgae strains that it has stable metabolic phenotype to improve the cumulative effect of target products [114,115]. However, random mutagenesis produces unpredictable effects, with large cost of investment and a potential for cancer in the environment and in humans. The ALE mechanism requires further exploring, and the characteristics of evolution are difficult to define. Genetic engineering is time-consuming, expensive, and limited by a deep understanding of the genetic mechanisms of microalgae [115,116]. Therefore, how to sift the appropriate species of microalgae and ameliorate it in order to obtain the commercial application of microalgae species need to be researched.

### 6.2. To Optimize the Culture Conditions of Microalgae

The differences in culture conditions (such as light, temperature, pH, nutrient elements, and $CO_2$ concentration) will change the production and $CO_2$ fixation efficiency of microalgae [117]. The viability of $CO_2$ fixation applications in microalgae is dominated by photosynthetic productivity and the biomass output of microalgae. During the growth of microalgae, carbon fixation efficiency is maximized by strictly controlling the growth conditions.

### 6.3. To Construct of Microalgae Carbon Dioxide Fixation Technology as the Core of the Industrial Model

By combining microalgae $CO_2$ fixation technology with wastewater treatment and flue gas treatment, a green and low-carbon recycling industry model with microalgae as the core is formed. Microalgae cultivation provides an effective, environmentally friendly,

and continuable solution for wastewater and flue gas treatment [95,118]. This scheme controls the emission of $CO_2$, fosters the recycling of nutrients in wastewater (e.g., nitrogen and phosphorus), absorbs pollutants in wastewater and flue gas, reduces the cost of $CO_2$ fixation, produces high-value-added products, and achieves win–win environmental and economic benefits [119,120]. However, most of the studies in the existing literature are conducted in laboratory conditions, and the follow-up studies should give priority to large-scale application to reasonably evaluate their industrial feasibility.

### 6.4. Comprehensive Development and Utilization of High-Value-Added Products of Microalgae

Microalgae biomass can be used extensively in a wide range of industries, which is crucial for alleviating the energy crisis, reducing environmental contamination, and promoting food security [115,121,122]. Improving the productivity of related products (such as lipids, pigments, and nutrients) is available for improving the economic benefits of microalgae $CO_2$ fixation technology [123,124], which contributes to impelling industrialization development of $CO_2$ fixation technology.

### 6.5. Potential for Carbon Fixation in the Microalgae

In addition to carbon sequestration in microalgae, there are some commonly used methods of carbon capture. Table 4 summarizes the advantages and limitations of various methods. Hence, microalgae $CO_2$ fixation technology has a wide range of application prospects as a new carbon capture technology. In the industrial field, it can be applied in the waste gas treatment of coal, chemical industry, and other industries, to convert $CO_2$ into useful microalgae biomass via photosynthesis [102]. Similarly, microalgae carbon sequestration technology can also be combined with wastewater treatment to purify water quality by absorbing nitrogen, phosphorus, and other nutrients in wastewater, which can be used in agriculture, industry, and other fields [125]. In the field of energy, biomass generated by microalgae $CO_2$ fixation technology can be used to produce biofuels, which can replace traditional petroleum energy and reduce dependence on fossil fuels, thus reducing $CO_2$ emissions. Microalgae biomass can also be used as a source of raw materials for medicines, cosmetics, food, and other fields [126]. There is still a long way to go to apply the achievements of carbon fixation technology from lab-scale research to large-scale application. For example, industrial carbon sequestration technology of microalgae is still faced with the problems such as low carbon sequestration efficiency, and further reduction in production costs. With the continuous progress of technology, the application scope of microalgae $CO_2$ fixation technology will continue to expand, the cost will continue to reduce, and the production efficiency will also continue to improve. Therefore, microalgae $CO_2$ fixation technology has great development potential and application prospects.

In short, microalgae carbon sequestration technology can reduce $CO_2$ emissions and delay global warming while providing high-value-added products to achieve $CO_2$ resource utilization. Therefore, compared with other carbon fixation methods, it has higher cost benefits, potential for effective carbon fixation, etc., which meets the requirements of green development and plays an important role in the realization of the goal of carbon neutrality.

**Table 4.** Comparison of various carbon capture methods.

| Category | Method | Description | Advantages | Limitations | Ref. |
|---|---|---|---|---|---|
| Physical | Geologic injection | Separate and capture $CO_2$, transport it to a storage location, and inject it deep underground for long-term isolation from the atmosphere | ♦ Take use of space available | ♦ High costs<br>♦ Special geological requirements | [127] |
| | Oceanic injection | Injection of $CO_2$ into deep ocean | ♦ Significant capacity to store $CO_2$ | ♦ High costs<br>♦ Low storage permanence<br>♦ Considerable ecological impacts | [128] |

**Table 4.** *Cont.*

| Category | Method | Description | Advantages | Limitations | Ref. |
|---|---|---|---|---|---|
| Chemical | Chemical absorption | Chemical absorption and desorption concept, determined by solubility of $CO_2$ | ♦ Environmentally safer<br>♦ Capture efficiency to 90% | ♦ Inefficient $CO_2$ capture capacity<br>♦ High evaporation loss of solvent<br>♦ Poor thermal stability,<br>♦ Equipment corrosion | [26] |
| | Mineral carbonation | $CO_2$ reacts with calcium- or magnesium-bearing rocks to form magnesite or calcite | ♦ $CO_2$ is converted to a solid substrate that can be reused as a building material or disposed of in surface facilities. | ♦ Need for a significant amount of reagent | [128] |
| Biological | Forest planting | Absorption of $CO_2$ through the photosynthesis of the trees | ♦ Significant social benefits<br>♦ Environmentally friendly | ♦ Large land area requirement<br>♦ Low carbon fixation efficiency compared to microalgae | [30] |
| | Microalgae carbon fixation | Carbon sequestration by microalgal photosynthesis | ♦ Environmentally friendly<br>♦ Economically feasible<br>♦ Plant in seawater or wastewater and require no a large area of soil resources<br>♦ $CO_2$ fixation more efficiently than land plants | ♦ Sensitive to living conditions (e.g., pH, toxic substances, and $CO_2$ concentrations)<br>♦ Improve the cost performance of the cultivation | [21] |

## 7. Conclusions

Due to global warming, microalgae carbon sequestration is among the most promising and sustainable methods to fix $CO_2$. Microalgae, with a relatively faster growth rate and higher $CO_2$ fixation efficiency, can also be used in wastewater treatment, flue gas treatment, biofuel production, as well as high-value-added products manufacture (e.g., proteins and lipids), which can maximize the economic benefits. Hence, it is an ideal material for biological capture of $CO_2$ under the background of "dual-carbon". Nevertheless, the future application of microalgae is still facing quite a few challenges. At present, many kinds of microalgae have been used for laboratory-scale testing. Limited by the technical level and production cost, the large-scale industrial biological sequestration of microalgae is still in its infancy, and no pilot-scale studies have yet been conducted.

To accelerate the realization of the carbon sequestration scale of microalgae, we should start by studying the microalgae $CO_2$ fixation mechanism, combine the $CO_2$ fixation technology of microalgae with high-value-added biomass production, promote the industrialization of microalgae $CO_2$ fixation technology, and establish and optimize a better mechanism, to achieve the purpose of promoting the circular economy and sustainable development. Based on this, carbon sequestration technology of microalgae needs to be innovated and improved according to the following aspects: (1) using relevant technologies to select suitable strains, (2) making the microalgae growing process more ecologically friendly by utilizing flue gas and wastewater treatment technology, and (3) to make full use of high-value-added products to achieve the greatest economic benefits. It is believed that the economically feasible and environmentally friendly industrialization model with microalgae $CO_2$ fixation technology as the core will be promoted and widely used in the foreseeable future.

**Author Contributions:** Conceptualization, G.L. and W.X.; methodology, G.L.; software, W.X.; validation, T.Y., W.X. and G.L.; formal analysis, T.Y.; investigation, W.X.; resources, G.L.; data curation, W.X.; writing—original draft preparation, W.X.; writing—review and editing, T.L.; visualization, W.X.; supervision, T.L.; project administration, T.L.; funding acquisition, G.L. All authors have read and agreed to the published version of the manuscript.

**Funding:** This research was funded by the National Natural Science Foundation of China (Grant NO. 32172277).

**Data Availability Statement:** The data presented in this study are available on request from the corresponding author.

**Acknowledgments:** We wish to thank the National Natural Science Foundation of China (32172277) and the Beijing Technology and Business University for their support.

**Conflicts of Interest:** The authors declare no conflict of interest.

## Appendix A

**Table A1.** Characteristics of the modified microalgal strains by ALE.

| Initial Strains | Medium | Initial Cell Density | Cycles | Stresses | Outcome | Ref. |
|---|---|---|---|---|---|---|
| *Chlorella* sp. | BG11 | $OD_{750}$ of 0.1 | 31 cycles (97 days) | 10% $CO_2$ | The maximal biomass concentration of AE10 was $3.68 \pm 0.08$ g/L in 30% $CO_2$, which was 2.94 fold compared to the original strain. | [70] |
| *Chlamydomonas reinhardtii* | TAP and TAP-N | $OD_{680}$ of 0.1 | 28 cycles (84 days) | Nitrogen starvation | ALE combined with nitrogen starvation substantially increased total lipid production, particularly for low-starch mutants. The endpoint strain of cc4334 under nitrogen starvation stress had the highest lipid productivity. | [129] |
| *Schizochytrium* sp. HX-308 | Main culture medium contained 40 g/L glucose and 0.4 g/L yeast extract | - | 40 cycles (40 days) | High oxygen | The adapted strain generated higher cell dry weight and lower lipid accumulation. | [90,130] |
| *Schizochytrium* sp. HX-308 | Medium with 30 g/L NaCl at a concentration of 1% *v/v* | - | 150 days | High salinity | The ALE150 showed a maximal cell dry weight of 134.5 g/L and a lipid yield of 80.14 g/L, representing a 32.7 and 53.31 increase over the starting strain, respectively. | [131] |
| *Chlorella* sp. L5 | TAP medium with 500 mg/L phenol | 0.6 g/L | 31 cycles (95 days) | High concentration phenol | The upregulations of the genes according to antioxidant enzymes and carotenoids synthesis were tolerated high phenol. | [132] |
| *Chlorella* sp. | TAP medium | 0.6 g/L | 31 cycles (95 days) | High concentration phenol | The strain had higher phenol biodegradation rates. | [133] |
| *Phaeodactylum tricornutum* (CCMP-2561) | Artificial seawater added with f/2 medium without silica | $1 \times 10^6$ cells/mL | 35 cycles (nearly 252 days) | Reducing salinity | 70% salinity potentiated the algae to enhance PUFAs. | [134] |
| *Picochlorum* sp. BPE23 | Liquid growth medium | $OD_{750}$ of 0.2 | 322 days | Supra-optimal temperature | At the optimal growth temperature of 38 °C, the biomass yield on light was 22.3% higher, and the maximal growth rate was 70.5% higher than the wild type. | [135] |
| *Synechocystis* sp. PCC 6803 | BG11 medium supplemented with 1.5% agar | $OD_{730}$ of 0.2 | 43 cycles (303 days) | 3% NaCl | All ALE-generated strains except S3 and S7 had a significantly higher growth rate than the control strain | [136] |

**Table A1.** *Cont.*

| Initial Strains | Medium | Initial Cell Density | Cycles | Stresses | Outcome | Ref. |
|---|---|---|---|---|---|---|
| *K. marxianus* | YPD medium | $OD_{600}$ of 0.8 | 65 days | Various temperature | The adapted *K. marxianus* strain accumulates glycerol and trehalose in response to lactose stress and ameliorate osmotolerance in *K. marxianus* cells. | [137] |
| *Scenedesmus* sp. SPP | Modified Chu13 medium | - | 10 days | Salinity stress, light stress, temperature stress | The triple stress-adapted strain showed the highest lipid content. | [138] |
| *Schizochytrium* sp. CCTCC M209059 | Main culture medium contained 40 g/L glucose and 0.4 g/L yeast extract | - | 80 days | High temperature | The adaptive strain showed a higher growth rate and lower temperature sensitivity. | [130,139] |
| *Nannochloroposis* oculata CCMP525 | The f/2 agar medium | 0.32 g/L | 24 cycles | High temperature | In a 2-L photobioreactor at 35 °C, biomass and lipid productivity were 1.43-fold and 2.24-fold higher, respectively, than wild type at 25 °C. | [140] |
| *Crypthecodinium cohnii* ATCC 30556 | Regular fermentation medium (27 g/L glucose, 25 g/L sea salt, and 6 g/L yeast extract) | $OD_{490}$ of 0.1 | 280 cycles (840 days) | Varying contents of the fermentation (30–90%) supernatant | The cell growth and DHA productivity of the evolved strain (FS280) were increased by 161.87 and 311.23%. | [77] |
| *Chlorella* sp. AE10 | BG11 medium | 0.04 g/L | 46 cycles (138 days) | High salinity | *Chlorella* sp. S30, has the potential for $CO_2$ capture under 30 g/L salt and 10% $CO_2$ conditions. | [141] |
| *Dunaliella salina* CCAP 19/18 | The f/2 medium | $1 \times 10^5$ cells/ml | 5 cycles (25 days) | Blue LED | The beta-carotene concentration (33.94 ± 0.52 μM) was enhanced by 19.7% compared to that observed for the non-ALE-treated wild type of D. *salina* under the B-R system (28.34 ± 0.24 μM). | [142] |

**Table A2.** Comparison of the removal efficiency of pollutants in different wastewater types by various microalgae.

| Waste Stream | Wastewater Source | Species | Reactor | Light Intensity | Ambient Temperature (°C) | Sampling Time (day) | Water Quality Index (mg/L) | | | Nutrient Removal Efficiency (%) | | | Maximum Biomass Concentration (g/L) | CO$_2$ Bio-Fixation (mg/L/d) | Lipid Productivity (mg/L/d) | Ref. |
|---|---|---|---|---|---|---|---|---|---|---|---|---|---|---|---|---|
| | | | | | | | COD * | TN * | TP * | COD | TN | TP | | | | |
| Agricultural wastewater | Soybean processing wastewater | *Chlorella* sp. L166 | Erlenmeyer flask | 6000 lux | 25 ± 1 | 18 | 5320 | 106.99 | 23.28 | 78.20 | 96.07 | 95.55 | 1.52 | - | - | [109] |
| | Starch processing wastewater | *Chlorella pyrenoidosa* | Conical flask | 127 µmol m$^{-2}$ s$^{-1}$ | 25 ± 1 | 6–7 | 702.4–1026.2 | 240.3–382.7 | 22.7–40.2 | 65.99 | 83.06 | 96.97 | 1.90 | - | 30.15 | [143] |
| | Swine wastewater | MBFJNU-1 | Flask | - | - | 12 | 824.53 | 547.78 | 81.72 | - | 90.51 | 91.54 | 0.63 | - | - | [144] |
| Agricultural wastewater and municipal wastewater | Swine wastewater 2:2 secondary treated municipal wastewater | *Chlorella sorokiniana* | Erlenmeyer flask | 126 µmol m$^{-2}$ s$^{-1}$ | 28 ± 2 | 10 | - | 337.32 | 30.88 | - | 63.90 | 93.02 | 1.31 | - | 23 | [145] |
| | Swine wastewater 1:3 secondary treated municipal wastewater | *Desmodesmus communis* | Erlenmeyer flask | 126 µmol m$^{-2}$ s$^{-1}$ | 28 ± 2 | 10 | - | 188.61 | 15.77 | - | 88.02 | 99.73 | 1.02 | - | - | |
| Municipal wastewater | Wastewater influent after primary settling tank | *Chlorella sorokiniana pa.91* | Erlenmeyer flask | 4000 lux | 30 | 16 | 211.4 | 2.01 (NO$_3^-$) 0.06 (NO$_2^-$) 34.1 (NH$_4^+$) | 6.1 (PO$_4^{3-}$) | 76 | 73 (NH$_4^+$) 93 (NO$_3^-$) | 83 (PO$_4^{3-}$) | 3.21 | - | - | [146] |
| | The primary sedimentation tank | *Chlorella vulgaris* ATCC 13482 | Cylindrical glass bottles | 90 ± 5 µmol m$^{-2}$ s$^{-1}$ | 25 | 10 | 293 | 46.67 | 19.50 | - | 93.40 | 94.10 | 0.94 | 140.91 | - | [147] |
| | | *Scenedesmus obliquus* FACHB 417 | Cylindrical glass bottles | 90 ± 5 µmol m$^{-2}$ s$^{-1}$ | 25 | 10 | 293 | 46.67 | 19.50 | - | 91.50 | 91.30 | 0.87 | 129.82 | - | |
| | Effluent of anaerobically digested food wastewater | *Scenedesmus bijuga* | Erlenmeyer flask | 80 µmol m$^{-2}$ s$^{-1}$ | 30 | 28 | 5923 | 2370 | 47.80 | - | 86.60 | 90.50 | 1.49 | - | 15.59 | [148] |
| | The sewage from the sewer | *Tetradesmus obliquus* PF3 | Conical flask | 6000 lux | 25 ± 1 | 5 | 267 | 43 | 4.9 | 90 | 93.20 | 99 | 1.8 | 551 | - | [149] |
| | Sterilized sewage | *Tetradesmus obliquus* PF3 | Conical flask | 6000 lux | 25 ± 1 | 5 | 210 | 40 | 4.70 | 42 | 94.70 | 99 | 1.8 | 558 | - | |
| | Targeting the tertiary treatment of wastewater | *Neochloris oleoabundans* | Erlenmeyer flask | 60 µmol·m$^{-2}$ s$^{-1}$ | 25 | 14 | - | 42 | 18.50 | - | 100 | 31.30 | 1.17 | 145 | - | [150] |
| | Domestic wastewater from secondary settling tanks | *Scenedesmus obliquus* | Conical flask | 14,500 lux | 23 ± 2 | 10 | 72.16 | 12.44 | 1.08 | - | 98.90 | 97.60 | - | - | - | [151] |
| | Manure wastewater | *Scenedesmus dimorphus* (FACHB-496) | Erlenmeyer flask | 60–80 µmol·m$^{-2}$ s$^{-1}$ | 26 ± 2 | 7 | - | 306.15 | 115.08 | - | 88.16 | 73.98 | - | 638.13 | - | [152] |
| Industrial wastewater | Artificial brewery wastewater | *Scenedesmus* sp. 336 | Erlenmeyer flask | 6000 lux | 25 ± 1 | 10 | 2100 | 45 | 7 | 73.66 | 75.96 | 95.71 | - | - | 38 | [153] |
| | | *Chlorella* sp. UTEX1602 | Erlenmeyer flask | 6000 lux | 25 ± 1 | 10 | 2100 | 45 | 7 | 44.97 | 81.43 | 97.54 | - | - | - | |
| | Palm oil mill effluent from an anaerobic treatment pond | *Chlorella* sp. (UKM2) | Transparent glass bottle | 266 µmol m$^{-2}$ s$^{-1}$ | 25 ± 2 | 15 | 2900 | 330 | - | - | 80.90 | - | - | 120.80 | - | [154] |
| | Waste molasses | *Scenedesmus* sp. Z-4 | Erlenmeyer flask | 3000 lux | 10 | 7 | 514,000 | 458 | 67 | 87.2 | 90.50 | 88.60 | 2.5 | - | 78 | [155] |
| | Membrane-treated distillery wastewater | *Chlorella vulgaris* | Erlenmeyer flask | 2000 lux | 25 | 9 | - | - | - | 72.24 | 80 | 94 | 0.65 | - | - | [156] |
| | Textile wastewater | Mixed microalgae (*Chlorella* Species and *Scenedesmus* sp.) | Conical flask | 212.77 mol m$^{-2}$ s$^{-1}$ | - | 13 | 1900 | 480.50 | 31 | 78.78 | 93.30 | 100 | 0.4 | - | - | [157] |
| | The simulated brewery effluent | *Scenedesmus obliquus* | Erlenmeyer flask | 12,000 lux | 30 ± 3 | 9 | 3635 | 54 | - | 57.5 | 20.80 | - | 0.9 | - | - | [158] |

* COD: chemical oxygen demand; TN: total nitrogen; TP: total phosphorus.

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
