# Peer review of "Optimization and Process Effect for Microalgae Carbon Dioxide Fixation Technology Applications Based on Carbon Capture: A Comprehensive Review"

_carbon, 2023_

Round 1

Reviewer 1 Report

Biosolids are essential for achieving carbon neutral technologies and this review examines microalgae carbon sequestration technologies.The effects of light, temperature, pH, nutrient and CO2 concentrations on the carbon sequestration capacity of microalgae are discussed.Strategies such as stochastic mutagenesis, adaptive laboratory evolution and genetic engineering are also evaluated.

There is definite scientific value and the article can be accepted with minor revision.

1. A technical flow chart needs to be drawn to illustrate the logical flow of the overview.

2. The conclusions are excessively brief and need to be broken down in detail, especially in the section on future technological developments.

3. Readers would probably prefer to see the future potential of the technology and the amount of carbon neutrality it can contribute to, and hopefully add to the discussion in this section.

4. The format of some references needs further modification, e.g. 2 in Ref. 72, CO2, is not listed in the table below.

Reviewer 2 Report

The manuscript with the title “Optimization and process effect for microalgae carbon dioxide fixation technology applications based on carbon capture: A comprehensive review” focused on the use of microalgae for carbon capture. The manuscript is well-written and suitable for a wide range of readers. However, there are a few concerns that need to be addressed before publishing. Below are my comments:

1. Line 175: “CA has a noticeable effect on catalyzing CO2 conversion”. Please include the factors that can affect CO2 conversion efficiency.

2. Line 152: “When CO2 is dissolved in fluid, HCO3−, CO32− H2CO3, are formed along with dissolved CO2”. Which one is preferred by algae and why? Also, what is the percentage of these compounds at a specific pH?

3. Please replace “PH” with “pH” in Line 153 and Figure 3.

4. If possible please include a table containing the species name, max operating temperature, and total conversion.

5. Section 4.1.2. Physical mutagenesis: “UV mutagenesis is the most common because of 323 flexible control, high safety, economy, fast and effective”. What is meant by high safety? UV is completely safe, or do we need to take some precautions? Further, is it applicable to all species to create mutagenesis?

6. Line 387: Please check the level of air CO2 at present. The current CO2 level in the atmosphere is way higher, around 416 ppm.

7 Line 392: “Flue gas is a cheap and abundant source of CO2, 400 times higher than the concentration in the atmosphere”. Again, this depends on whether it is a coal flue gas or natural gas flue gas. Please refer to this reference https://doi.org/10.1007/978-981-32-9804-0_13.

8. What are the author’s views on the cost and efficiency while using algae for CO2 Capture?

Reviewer 3 Report

This work reviews the process of CO2 fixation using microalgae. The work is very comprehensive and very well organized. I think this paper will be of interest to the readers with considering the following minor suggestions. 

1. It would be interesting if it possible to provide a direct comparison between CO2 uptake by microalgae ponds and other approaches like direct air capture (section 5.1). 

2. I also suggest that the others add a section discusses the advantages and limitations (e.g., carbon capture capacity, regeneration, cost, etc.) of microalgae approach over other pathways such as direct carbon capture and carbon mineralization. 
